# Alpha-Lipoic Acid Plays a Role in Endometriosis: New Evidence on Inflammasome-Mediated Interleukin Production, Cellular Adhesion and Invasion

**DOI:** 10.3390/molecules26020288

**Published:** 2021-01-08

**Authors:** Fiorella Di Nicuolo, Roberta Castellani, Alessandra De Cicco Nardone, Greta Barbaro, Carmela Paciullo, Alfredo Pontecorvi, Giovanni Scambia, Nicoletta Di Simone

**Affiliations:** 1Istituto Scientifico Internazionale PaoloVI, ISI, Università Cattolica del Sacro Cuore, 00168 Rome, Italy; fiorella.dinicuolo@gmail.com; 2Istituto di Clinica Ostetrica e Ginecologica, Università Cattolica del Sacro Cuore, 00168 Rome, Italy; roberta.castellani@unicatt.it (R.C.); greta.barbaro@libero.it (G.B.); carmela.paciullo.1989@gmail.com (C.P.); 3U.O.C. di Ostetricia e Patologia Ostetrica, Dipartimento di Scienze della Salute della Donna, del Bambino e di Sanità Pubblica, Fondazione Policlinico Universitario A. Gemelli IRCCS, 00168 Rome, Italy; alessandra.decicco@policlinicogemelli.it; 4Dipartimento di Scienze Gastroenterologiche, Endocrino-Metaboliche e Nefro-Urologiche, Fondazione Policlinico Universitario A. Gemelli IRCCS, 00168 Rome, Italy; alfredo.pontecorvi@policlinicogemelli.it; 5U.O.C. Ginecologia Oncologica, Dipartimento per la Salute della Donna e del Bambino e della Salute Pubblica, Fondazione Policlinico Universitario A. Gemelli IRCCS, 00168 Rome, Italy; giovanni.scambia@policlinicogemelli.it

**Keywords:** α-lipoic acid, endometriosis, NALP-3 inflammasome, estrogen receptor (ER) β

## Abstract

Endometriosis is an estrogen-linked gynecological disease defined by the presence of endometrial tissue on extrauterine sites where it forms invasive lesions. Alterations in estrogen-mediated cellular signaling seems to have an essential role in the pathogenesis of endometriosis. Higher estrogen receptor (ER)-β levels and enhanced ER-β activity were detected in endometriotic tissues. It is well known that ER-β interacts with components of the cytoplasmic inflammasome-3 (NALP-3), the NALP-3 activation increases interleukin (IL)-1β and IL-18, enhancing cellular adhesion and proliferation. Otherwise, the inhibition of ER-β activity suppresses the ectopic lesions growth. The present study aims to investigate the potential effect of α-lipoic acid (ALA) on NALP-3 and ER-β expression using a western blot analysis, NALP-3-induced cytokines production by ELISA, migration and invasion of immortalized epithelial (12Z) and stromal endometriotic cells (22B) using a 3D culture invasion assay, and matrix-metalloprotease (MMPs) activity using gelatin zymography. ALA significantly reduces ER-β, NALP-3 protein expression/activity and the secretion of IL-1β and IL-18 in both 12Z and 22B cells. ALA treatment reduces cellular adhesion and invasion via a lower expression of adhesion molecules and MMPs activities. These results provide convincing evidence that ALA might inhibit endometriosis progression.

## 1. Introduction

Endometriosis is a chronic estrogen-dependent gynecological disease, it is defined by the attachment of endometrial tissue on extrauterine sites, where it forms invasive lesions. It affects ~5–15% of all fertile women in the reproductive age and 20–50% of all infertile women [1,2].

This disease is commonly associated with severe clinical symptoms, such as chronic pelvic pain, dyspareunia, infertility, and dysmenorrhea, leading to a high social cost because of missed working days, surgical operations and the frequent seeking of assisted reproduction technologies [3,4,5].

Nevertheless, the pathogenesis of endometriosis is still not clearly defined, and over time, several etiological theories have been pointed out, such as retrograde menstruation, coelomic metaplasia, or lympho-vascular metastasis [6]. A more recent hypothesis is that women that develop endometriosis have a defective immune system unable to recognize the endometrial fragments within the pelvic cavity [7,8].

New studies suggest that inflammation plays a key role in pathogenesis of endometriosis, from the implantation to proliferation and migration of endometriotic cells. The overexpression of cytokines such as TNF-α, interleukins, TGF-β, monocyte chemoattractant protein-1 (MCP-1) and the resulting chronic inflammation have been shown to contribute to chronic pelvic pain and endometriosis-related infertility [9,10,11,12,13].

In a recent research, Han et al. showed that estrogen receptor β (ER-β), which is remarkably high in endometriotic tissues, blocked TNF-α, induced cellular apoptosis and enhanced inflammasome NALP-3-mediated IL-1β production, involved in the adhesion and proliferation of endometrial cells [14].

Inflammasomes are cytosolic multi-protein complexes in charge of the initiation of inflammatory process during innate immune response [15]. The NLR-family pyrin domain-containing protein 3 (NALP-3) inflammasome is the most characteristic complex. Its activation triggers the cleavage of pro-interleukin (IL)-1β and pro-IL-18, which are mediated by caspase-1, and secretes mature forms of these mediators from cells to promote the further inflammatory process and oxidative stress [16].

The importance of NALP-3 in the endometriosis pathogenesis was highlighted by the observation that ectopic endometrial lesion volumes were greatly reduced in a model of NALP-3-deficient mice compared to controls [14]. These data suggest that ER-β might protect endometriotic cells from immunosurveillance, by inhibiting TNF-α-induced cell death pathways and may promote ectopic lesions growth by enhancing NALP-3-induced IL-1β production [17]. In any case, NALP-3 plays a critical role in a lot of inflammation-mediated human diseases and represents a promising drug target for novel anti-inflammatory therapies.

Recently, we conducted an investigation on endometrial NALP-3 in women with idiopathic recurrent pregnancy losses and we observed that α-lipoic acid (ALA) played an inhibitory effect on endometrial NALP-3 expression/activation [18,19]. ALA, also known as 1,2-dithiolane-3-pentanoic acid or thioctic acid, is a five-carbon carboxylic acid bound to a five-atom cyclic disulfide. It has two enantiomeric configurations (*R*-ALA and *S*-ALA) and its reduced form is known as dihydrolipoic acid (DHLA) (Figure 1) [20]. Both *S* and *R* enantiomers are present equally in ALA; however, the *R* isomeric form is present naturally, while the *S* isomer is prepared through chemical processes. Foods are a natural source of the *R* enantiomer, naturally produced inside living organisms forming covalent bonds with proteins. While ALA exists in nature as *R* enantiomer, synthetic supplementation consists of a racemic composition of *R* and *S* forms [21,22].

ALA’s pharmacological effects are primarily related with its antioxidant activity, but ALA and DHLA have also demonstrated interesting cardiovascular, anti-ageing, detoxifying, anti-inflammatory, anti-cancer, and neuroprotective properties [23].

ALA reduces the pro-inflammatory cytokine levels, such as TNF-α, IL-1β, -6, -8 and -17, interferon (INF)-γ as well as the production of Vascular and Intercellular cell adhesion protein (VCAM-1 and ICAM-1). It is also widely known as an inhibitor of NF-KB and it is able to inhibit NF-kB-dependent expression of metalloproteinase-9 in vitro [24,25,26].

In addition, ALA induces the release of the anti-inflammatory cytokine IL-10 [27]. It is known that ALA suppresses the number and percentage of T helper (Th)1 and Th17 cells and the Natural Killer (NK) cell cytotoxicity [22]. It increases the splenic T-regulatory (T-reg) cells, involved in fighting excessive inflammation [22]. Taking into consideration all these observations, ALA seems to strongly contribute to counteract inflammatory conditions [22,23,24,25,26,27].

In light of all of these data, the present study aims to investigate the potential effect of ALA on ER-β expression and NALP-3-induced cytokines production in immortalized endometriotic cells, as well as the underlying molecular mechanisms. It is evident that the knowledge of the molecular mechanisms involved in the pathogenesis of endometriosis may be valuable for therapeutic purposes such as the identification of potential new molecular targets for pharmacological intervention.

Our results provide further insights regarding the ability of ALA to inhibit endometriotic cellular ICAM-1-mediated adhesion and MMP-mediated cellular invasion, suggesting a role on the inhibition of endometriosis progression.

## 2. Results

### 2.1. Alpha Lipoic Acid Effects on Endometriotic Cells Viability

MTT assay showed that ALA treatment (0–1.0–10 mM) did not affect cell viability in both 12Z and 22B cell lines after 24 h incubation. (Table 1).

### 2.2. Alpha-Lipoic Acid Regulates ER-β and NALP-3 Proteins

We evaluated the role of ALA on endometriotic ER-β and NALP-3 inflammasome protein expressions. The western blotting results showed that ALA treatment leads to a significant decrease in ER-β (Figure 2A,C) and in NALP-3 proteins (Figure 2B,D) in both 12Z and 22B cells. The ER-β and NALP-3 protein expressions were significantly lower in 12Z cells treated with ALA (1.0 mM; Figure 2A,B) than in 22 B cells. A higher dose of ALA (10 mM) significantly reduced ER-β and NALP-3 protein levels in both 12Z and 22B cells (Figure 2C,D, compared to baseline (controls: CTR).

### 2.3. Cytokine Levels

It has been demonstrated that the inflammasome complex regulates the activation of pro-inflammatory cytokines, IL-1β and IL-18, from inactive precursors to active molecules. As a consequence, we investigated the effect of ALA (1.0–10 mM) on the cytokines production in endometriotic cell lines: 12Z and 22B. We detected the levels of cytokines in cell lysates using an ELISA. As shown in Figure 3, reduced levels of both pro-inflammatory cytokines, IL-1β and IL-18 were found in 12Z cells (ALA 1.0 and 10 mM; Figure 3A,B). The treatment of the endometriotic stromal cell line, 22B, with ALA (1.0 mM) significantly reduced the production of IL-18 compared to control (CTR; Figure 3D). ALA did not affect IL-1β secretion in 22B cells.

### 2.4. Alpha Lipoic Acid Inhibits Endometriotic Cell Adhesion and Invasion

Further experiments were performed on 12Z and 22B cells adhesion and invasion. Endometriotic cells were treated with ALA for 24 h; the transwell assay revealed that ALA leads to a significant reduction of cell adhesion and invasion (Figure 4 and Figure 5). The ALA-induced inhibition of cell adhesion and invasion were more evident in epithelial endometriotic cell lines (12Z) than in the stromal cells (22B). The 12Z adhesion and invasion were reduced by ALA starting from 1 mM. Only a higher dose of ALA (10 mM) significantly reduced adhesion and invasion in 22B cells, compared to untreated cells (controls: CTR).

Subsequently, we investigated if intercellular adhesion molecule-1 (ICAM-1) and matrix metalloproteases (MMP-2 and -9) were involved in ALA-mediated inhibition of cellular adhesion and migration, respectively.

ICAM-1 is a well-known cell surface protein able to mediate the adhesion between cells and extracellular matrix. As shown in Figure 6, ALA (1.0–10 mM) inhibited the ICAM-1 protein expressions in both 12Z and 22B endometriotic cells compared to controls (CTR).

It is well known that extracellular matrix degradation by MMPs is required for cell migration and invasion. MMP-2 and MMP-9 are the major proteinases enabling the metastatic ability of endometriotic cells to infiltrate the basement membrane. Thus, we examined the role of the MMP-2 and MMP-9 in ALA-regulation of endometriotic cells invasiveness. The cells were treated with ALA (0–1.0–10 mM) and MMP-2 and -9 levels were investigated using gelatin zymography in the culture medium, as described in the ‘Materials and Methods’ section. In Figure 7, the ALA-reduced MMP-9 and MMP-2 expression is shown in both endometriotic cell lines, confirming the ALA role in regulating MMP activity.

## 3. Discussion

In the present paper we have demonstrated the role of ALA on the inhibition of endometriosis progression. In fact, on endometriosis cell lines (12Z and 22B), ALA significantly reduced ER-β protein levels and NALP-3 expression and activity. Even cellular adhesion and invasion appear to be inhibited, in the presence of ALA, due to a lower expression of adhesion molecules like ICAM-1 and a reduced matrix-metalloproteases (MMP-9 and MMP-2) activity.

It is known that the expression of ER-β is considerably high in endometriotic tissue compared to the normal endometrial tissue [28]. ER-β overexpression is related with an increased endometriotic lesion volume because ER-β increases cellular proliferation and decreases cellular apoptosis [28]. Furthermore, the importance of inflammasome (NALP) in the endometriosis pathogenesis is highlighted by the observation that ectopic endometriotic lesion volume is greatly reduced in NALP-3-deficient mice compared to non-deficient mice. Han et al. have recently demonstrated that the ER-β interaction with components of the inflammasome increases mature IL-1β and IL-18 production [14]. IL-1β is involved in the adhesion and proliferation of endometrial cells and ER-β-deficient cells show a low caspase-1 activity. As a consequence, there are low levels of IL-1β and IL-18. All these data suggest that ER-β promotes ectopic endometrial lesion growth by enhancing NALP-induced IL-1β production.

Interleukin-1β is a proinflammatory cytokine involved in nearly all events associated with the activation and regulation of inflammation [29,30,31]. It induces the release of other cytokines such as TNF-α, IL-6, and IL-1α as well as IL-1β itself along with other important factors involved in the growth and differentiation of immune cells (dendritic cells, neutrophils, lymphocytes). Interleukin-1β is essential for innate and adaptive immunity, taking part in B, T, and NK cell recruitment and in antigen presentation [32]. The other cytokine related to NALP-3 activation IL-18 is implicated in the differentiation of type 1 helpers and cytotoxic T lymphocytes, in the production of Th2 cytokines by T cells, basophils, NK cells and mast cells [33,34]. It is now commonly acknowledged that the activation and release of IL-1β and IL-18 require two distinct parallel signals [32,33,34]. The classification of these signals in vivo, in infections, or in inflammatory responses, is not fully known. In vitro studies imply that one signal is activated by pathogen-associated molecular patterns through toll-like receptor activation, conducting the synthesis of pro-ILs. The other signal is stimulated by several external stimuli (including external ATP, considered a hazardous signal) and causes the inflammasome assembly and the caspase-1 activation. Nevertheless, the accurate mechanisms through which upstream pathways lead to the formation of pro-ILs and recruitment of inflammasome is, to date, uncertain [35,36]. The overexpression of cytokines and a change in immune cell functions (in which macrophages appear to be the principal performers) seem to play an essential role in endometriosis, leading to chronic inflammation [5,6,7,8]. This condition may be, to some degree, accountable for consequent pelvic adhesions and pain.

It is clear that understanding the molecular mechanisms implicated in the pathogenesis of a chronic disease such as endometriosis may be relevant for therapeutic purposes, such as identifying potentially novel molecular targets for pharmacological treatment. Indeed, in line with the recent guidelines that surgical treatment must be addressed to extremely particular cases or consequently to the failure of medical treatment, the pharmacological plan must be taken into account as first-line therapy in all the remaining conditions.

Our research has confirmed the presence of ER-β as well as the NALP-3-inflammasome expression and the secretion of IL-1β and IL-18 in 12Z (epithelial) and 22B (stromal) endometriotic cell lines, and we have demonstrated how ALA is able to reduce ER-β protein levels as well as NALP-3 activity.

In order to further understand the pathophysiology of endometriosis and enable the research for its treatment, it is essential to gain insight into the mechanisms involved in the adhesion and invasion of endometrial cells. The main focus was defining the potential benefits that the administration of ALA might have on endometriotic cell matrix adhesion and invasion, both necessary for endometriotic implantation. We observed that the cellular adhesion and invasion are reduced in the presence of ALA via an inhibition in the expression of an adhesion molecule (such as ICAM-1) and MMPs activities.

Previous studies already demonstrated that IL-1β induced the expression of MMPs, via NF-KB activation [37]. Some types of MMPs, produced in response to IL-1β, are important proteolytic factors in the degradation of extracellular matrix [38]. Therefore, substances that inhibit the expression of MMPs might be applied as new therapeutic strategies to inhibit cell invasiveness.

It is known that ALA anti-inflammatory effects are exerted through the modulation of NF-kB [26]. Recently, Fratantonio et al. showed that nanomolar concentrations of oxidized ALA protect human endothelial cells (HUVEC) from TNF-α induced dysfunction, inhibit NF-κB activation, and block apoptosis following the activation of Nrf2 transcription factor [39]. As also demonstrated by Ying and coworkers, the pretreatment with ALA, but not with other tested antioxidants, inhibits TNFα-induced NF-κB activation and VCAM-1 and COX2 expression in HUVECs, confirming that the effects elicited by ALA are independent of its antioxidant function [40].

Since it has been reported that ALA is able to inhibit NF-KB-dependent expression of metalloproteinase-9 in vitro [26], we have examined the effect of ALA on MMPs secretion. We showed that ALA significantly reduced the production, NALP-3-dependent, of both MMP-2 and -9. These results provide convincing evidence that ALA might inhibit endometriosis progression by down-regulation of ICAM-1-mediated adhesion and MMP-mediated cellular invasion.

## 4. Materials and Methods

### 4.1. Cell Cultures

Immortalized human endometriotic epithelial and stromal cells (12Z and 22B, respectively) were maintained in Dulbecco’s modified Eagle’s medium (DMEM)/F12 medium supplemented with 10% fetal bovine serum, 100 U/mL penicillin G, and streptomycin (LifeTechnologies, New York, NY, USA) in a humidified atmosphere of 5% CO_2_–95% air at 37 °C.

The cell lines, isolated from active peritoneal endometriotic lesions, show characteristics of the active phase of endometriosis and thus are suitable for studying cellular and molecular behavior of endometriosis [41].

### 4.2. Pharmacological Treatments

After culturing overnight (37 °C, 5% CO_2_), cells were starved in fresh medium containing ALA (0–1.0–10 mM) and cultured (37 °C, 5% CO_2_) for 24 h. ALA (Thioctic Acid, Barenz Service SpA, Paderno Dugnano, Milan, Italy) was prepared by dissolving in absolute ethanol, and stock sample was further diluted in culture medium. The final concentration of absolute ethanol used in all of the experiments was 0.1%. The results from the treated cells were compared with the non-treated cell exposed to the 0.1% final concentration of absolute ethanol (controls: CTR).

To deplete endogenous steroids in the ERβ expression studies, the medium was changed to phenol red-free medium containing 10% charcoal/dextran stripped FBS (ThermoFisher Scientific, Rockford, IL, USA) for 24 h before ALA treatment.

Treated cells and supernatants were collected and stored at −80 °C until use in the western blot assay and ELISAs.

### 4.3. Cell Viability Assay

Cell viability was examined using a MTT assay (Vybrant^®^ MTT Cell Proliferation Assay Kit, Thermo Fisher Scientific, Rockford, IL, USA). Endometriotic cells (12Z and 22B) were seeded in a 96-well plate and treated with medium containing ALA (0–1.0–10 mM) for 24 h. Then, 10 μL MTT (12 mM) was added to the medium for 4 h. Next, the supernatant was discarded, and 50 μL DMSO was added to each well for 10 min at 37 °C and the optical density (O.D.) was measured at absorbance at wavelengths of 540 nm (Titertek Multiscan plus Mk II plate reader, ICN Flow Laboratories, Irvine, CA, USA).

### 4.4. SDS–PAGE and Immunoblotting

For western blotting, total cellular lysates were separated by 10% SDS-PAGE electrophoresis under reducing conditions and transferred (wet transfer method for blotting, 120 Volts, 1 h) to PVDF membranes (Millipore). The membranes were blocked at room temperature for 1 h in 5% non-fat dry milk, and incubated overnight at +4 °C with a specific primary antibody (Human NALP-3 antibody monoclonal mouse IgG, 1:1000, Clone CL0210, Novus Biologicals; anti –ERβ, Human ERβ/NR3A2 Antibody Monoclonal Mouse IgG1, 1:1000, Clone 733930, R&D Systems, Bio-Techne Ltd., Abingdon, UK). The membranes were washed with PBST and incubated in specific horseradish peroxidase-conjugated IgG diluted 1:2000 in 5% non-fat dried milk in PBST. A bound secondary antibody was detected by chemiluminescence. Bands were analyzed using the enhanced chemiluminescence (ECL™, Amersham, UK) by chemiluminescence imaging system Alliance 2.7 (UVITEC, Cambridge, UK) and quantified by the software Alliance V1607.

As positive load control cytosolic, the β-actin band (42-kDa) was detected by a mouse monoclonal anti-human β-actin antibody, 1:5000 (Sigma-Aldrich, St. Louis, MO, USA).

### 4.5. IL-18 and IL-1β Immunoassays

IL-18 and IL-1β levels were measured in supernatants obtained from 12Z and 22B cells (1 × 10^5^ well/ 24 well plate) treated with ALA (0–10 mM) for 24 h using an enzyme-linked immunoassay (ELISA) according to the manufacturer’s instructions (USCN Life Science Inc. and Cloud-Clone Corp. Houston, TX, USA). Briefly, samples or standards (100 µL) were added to each well coated with human monoclonal IL-18 or IL-1β antibodies. After 2 h of incubation at 37 °C, wells were washed and incubated with a specific enzyme-linked polyclonal antibody: horseradish peroxidase. Then, tetramethyl-benzidine substrate solution was added to each well, and the color developed in proportion to the amount of the proteins bound in the initial step. The plate was read on a Titertek Multiscan plus Mk II plate reader (ICN Flow Laboratories, Irvine, CA, USA), measuring the absorbance at wavelengths of 450 nm. IL-18: Detection Range 15.6–1000 pg/mL. Sensitivity: The minimum detectable dose of IL-18 is typically less than 5.9 pg/mL. IL-1β: Detection Range 15.6–1000 pg/mL. Sensitivity: The minimum detectable dose of IL-1β is typically less than 5.7 pg/mL.

### 4.6. Cell Adhesion Assay

Cell adhesion was measured using the Vybrant™ Cell Adhesion Assay Kit according to the manufacturer’s instructions (Thermo Fisher Scientific, Rockford, IL, USA). The cells, treated with ALA (0–10 mM) for 24 h, were loaded with calcein acetoxymethyl ester AM (5 µM) at 37 °C for 30 min. Calcein AM is non fluorescent, but, once loaded into cells, is cleaved by endogenous esterases to produce highly fluorescent calcein. Then, 100 μL of the calcein-labeled cell suspension was added to prepared microplate wells with ECM proteins, which was then incubated at 37 °C for 120 min. Non-adherent calcein-labeled cells were removed by careful washing, and the fluorescence was measured using a microplate reader (absorbance maximum of 494 nm and emission maximum of 517 nm, Berthold Technologies GmbH & Co. KG, Bad Wildbad, Germany).

### 4.7. I-CAM Immunoassay

I-CAM levels were measured in total lysates obtained from 12Z and 22B cells (2 × 10^5^ well/24 well plate) treated with ALA (0–10 mM) for 24 h, using an enzyme-linked immunoassay (ELISA, USCN Life Science Inc. and Cloud-Clone Corp. Houston, TX, USA) according to the manufacturer’s instructions as described above. I-CAM: Detection Range 15.6–1000 pg/mL. Sensitivity: The minimum detectable dose of this kit is typically less than 5.9 pg/mL.

### 4.8. Cell Invasion Assay

To assess the effect ALA on the invasiveness of endometriotic cells, we used a 3D culture invasion assay (Cultrex^®^ 3D Spheroid Cell Invasion Assay, Trevigen, Gaithersburg, MD, USA) according to the manufacturer’s instructions. This kit uses a 3D culture-qualified 96-well spheroid formation plate alongside a specialized spheroid formation extracellular matrix to drive the aggregation and/or spheroid formation of cells. Upon completion of spheroid formation, the spheroid is embedded in an invasion matrix composed of basement membrane proteins. This matrix forms a hydrogel network on which invasive cells can spread. At this point, invasion-modulating agents can be applied to the system to evaluate the impact on the cell response. In our experiments, spheroids of 12Z or 22B cells (3 × 10^3^/50 µL of cell suspension per well) were incubated with ALA (0–10 mM) for six days. Cell invasion was then visualized microscopically and quantified through image analysis software (ImageJ 1.46r, National Institute of Health).

### 4.9. Gelatin Zymography

MMP-2 and -9 levels in the supernatant of 12Z and 22B cells were measured using gelatin zymography. Cells (2 × 10^5^ well/24 well plate in serum free medium) were treated with ALA (0–10 mM) for 24 h. Samples were placed on SDS polyacrylamide gel containing 0.1% gelatin. Following electrophoresis, gels were washed three times for 10 min at room temperature in 2.5% Triton X-100 to remove the SDS. After overnight incubation at 37 °C in 50 mM Tris-HCl, pH 7.4, containing 5 mM CaCl_2_, 0.15 M NaCl, and 0.02% NaN_3_, gels were stained with 0.5% coomassie brilliant blue for 30 min and destained in 20% methanol and 10% acetic acid. Gelatinolytic activities were observed as a clear band of digested gelatin on a blue background. Images were acquired with a digital camera (Nikon, Tokyo, Japan), and bands were analyzed on the Image Analysis System Gel Doc 200 System (Bio-Rad Laboratories, Milan, Italy) using Quantity One quantitation software.

### 4.10. Statistical Analysis

Each experiment was repeated independently at least three times in duplicate. The results are presented as the mean ± S.E.M. The data were analyzed using one-way analysis of variance (ANOVA) followed by a post–hoc test (Bonferroni test). Statistical significance was determined at *p* < 0.05.

## 5. Conclusions

Our findings provided new evidence: ALA has the ability to inhibit ER-β expression, NALP-3-induced cytokines production, migration, and invasion in endometriotic cell lines. We studied human immortalized endometriotic cell lines derived from peritoneal lesions. Both cell lines share several phenotypic and molecular characteristics of primary cultured endometriotic cells and several data indicate that 12Z and 22B cells are a potential model system to study the progressive phase of endometriosis [41]. Therefore, more studies using primary stromal and epithelial cells isolated from endometriotic tissues are needed to confirm protective effect of ALA on endometriosis progression.

## Figures and Tables

**Figure 1 molecules-26-00288-f001:**
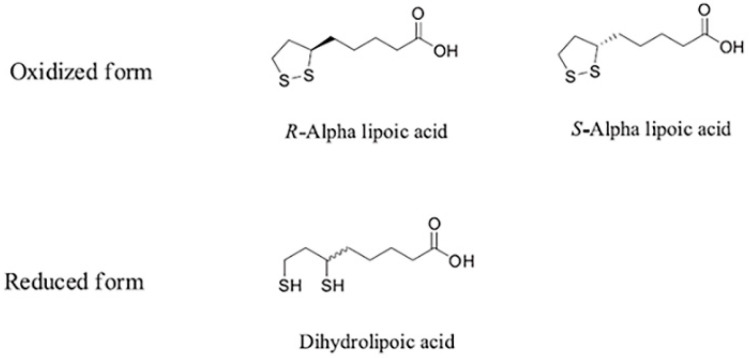
Chemical structure of alpha-lipoic acid (*R* and *S*) and dihydrolipoic acid. The alpha lipoic acid (ALA) is a five-carbon carboxylic acid bound to a five-atom cyclic disulfide. It has two enantiomeric configurations (*R*-ALA and *S*-ALA) and its reduced form is known as dihydrolipoic acid (DHLA).

**Figure 2 molecules-26-00288-f002:**
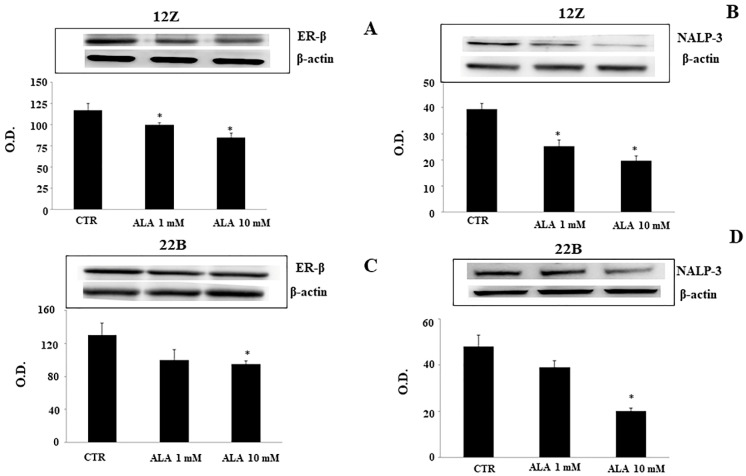
Alpha lipoic acid (ALA) effect on endometriotic ER-β and NALP-3 expression. Immortalized human endometriotic epithelial and stromal cells (12Z and 22B, respectively) were incubated with ALA (0–1.0–10 mM) and analyzed 24 h after treatment. ER-β (**A**,**C**) and NALP-3 levels (**B**,**D**) were quantified using a western blot analysis; ER-β and NALP-3 protein expression levels were compared to the constant level of β actin (loading control) and expressed as O.D. (Optical Density, means ± SEM of three independent experiments). * Statistical significance versus control cells (CTR): *p* < 0.01.

**Figure 3 molecules-26-00288-f003:**
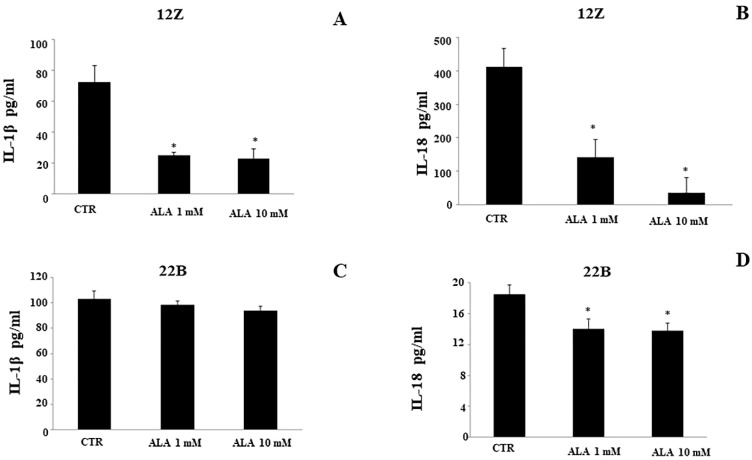
Alfa-lipoic acid regulates interleukins secretion. Immortalized human endometriotic epithelial and stromal cells (12Z and 22B, respectively) were incubated with ALA (0–1.0–10 mM) and the culture media were analyzed 24 h after treatment. IL-1β (**A**,**C**) and IL-18 (**B**,**D**) were examined using a colorimetric ELISA assay. Results were means ± SEM of three independent experiments and were expressed as pg/mL of protein levels. (* Statistical significance versus untreated cells (CTR): *p* < 0.01).

**Figure 4 molecules-26-00288-f004:**
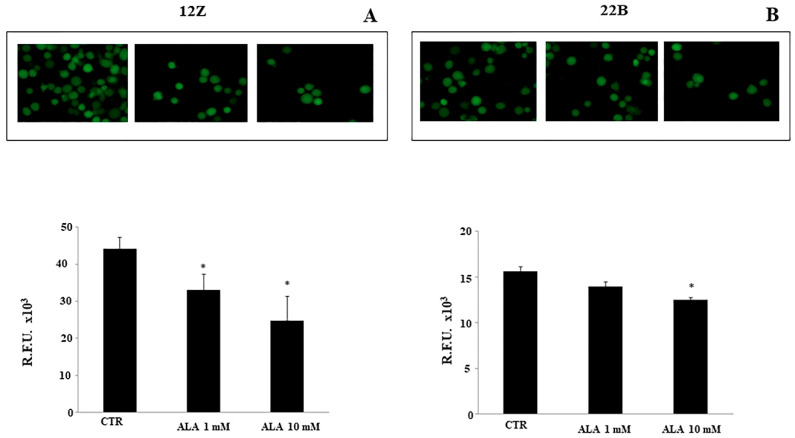
Alpha lipoic acid reduces endometriotic cell adhesion. After endometriotic cells were treated with ALA (0–1.0–10 mM) for 24 h, the transwell assay revealed that ALA inhibited the adhesion of both 12Z (**A**) and 22B (**B**) cells in a dose-dependent manner. The ALA-induced inhibition of cell adhesion was more evident in the epithelial endometriotic cell line (12Z) than in the stromal cells (22B). The 12Z adhesion was reduced in the presence of ALA 1.0 mM. Only a higher dose of ALA (10 mM) significantly reduced adhesion in 22B cells, compared to control cells (CTR). Results are means ± SEM of six independent experiments. * Statistical significance versus untreated cells (CTR): *p* < 0.05.

**Figure 5 molecules-26-00288-f005:**
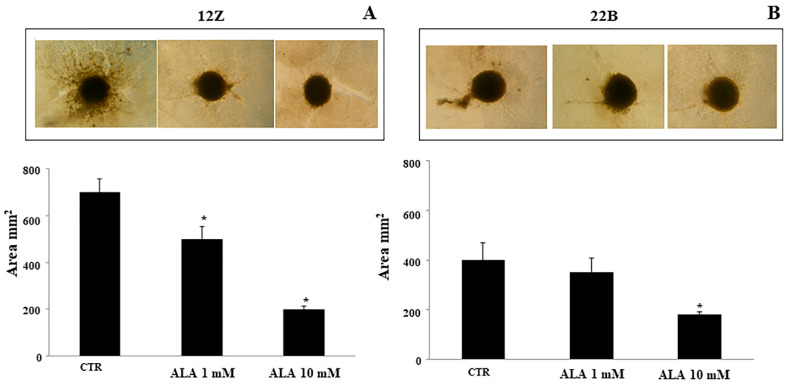
Alpha lipoic acid reduces endometriotic cell invasion. To assess the effect of ALA on the invasiveness of endometriotic cells, we used a 3D culture invasion assay. The ALA-induced inhibition of cell invasion was more evident in the 12Z epithelial endometriotic cell line (**A**) than in the 22B stromal cells (**B**). The 12Z invasion was reduced in the presence of ALA 1.0 mM. Only a higher dose of ALA (10 mM) significantly reduced invasion in 22B cells, compared to control cells (CTR). Results are means ± SEM of four independent experiments. * Statistical significance versus untreated cells (CTR): *p* < 0.05.

**Figure 6 molecules-26-00288-f006:**
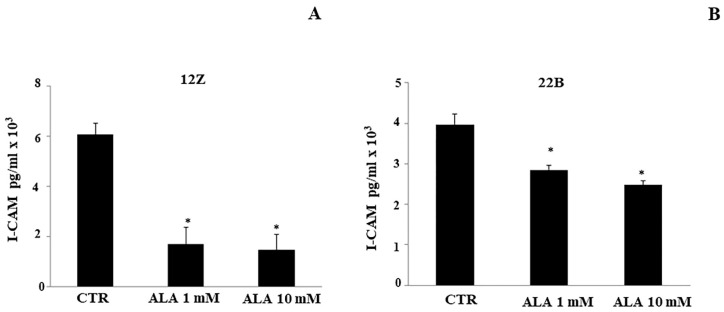
Alpha lipoic acid reduces endometriotic ICAM-1 protein levels. I-CAM levels were measured in total lysates obtained from 12Z (**A**) and 22B (**B**) cells treated with ALA (0–1.0–10 mM) for 24 h, using ELISA. As shown, ALA (1.0 and 10 mM) effectively inhibited the ICAM-1 protein levels of 12Z and 22B endometriotic cells compared to controls. Results are means ± SEM of three independent experiments. * Statistical significance versus untreated cells (CTR): *p* < 0.05.

**Figure 7 molecules-26-00288-f007:**
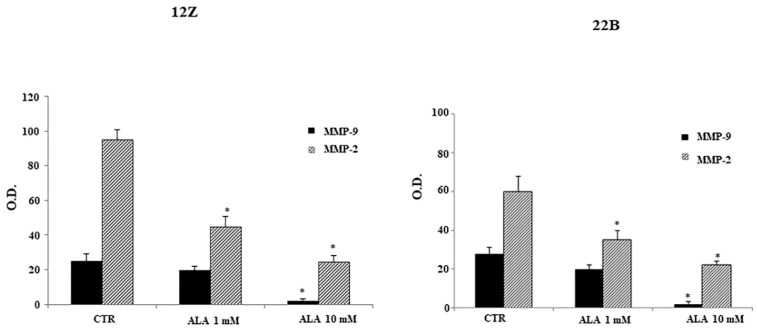
Alpha lipoic acid reduces MMP-2 and MMP-9 activities. Immortalized human endometriotic epithelial (12Z) and stromal cells (22B) were treated with ALA (1.0 and 10 mM) for 24 h. MMP-2 and -9 activities in the supernatant were measured using gelatin zymography as described in the ‘Materials and Methods’ Section. Figure shows that ALA reduced MMP-2 and MMP-9 activities in both 12Z and 22B endometriotic cells compared to controls (CTR). Results are means ± SEM of three independent experiments and were expressed as O.D. % of untreated cells. * Statistical significance versus untreated cells (CTR): *p* < 0.05.

**Table 1 molecules-26-00288-t001:** Viability of 12Z and 22B endometriotic cell lines.

	12Z	*p* Value	22B	*p* Value
ALA (mM)				
0	100.0 ± 1.6		100.0 ± 1.0	
1.0	93.4 ± 3.8	NS	96.8 ± 1.2	NS
10.0	98.0 ± 2.0	NS	90.9 ± 4.9	NS

Cell viability of epithelial (12Z) and stromal (22B) endometriotic cells treated with increasing concentrations of ALA (0–10 mM) for 24 h. The control condition (0 mM) was arbitrarily set at 100% and values are expressed as the mean ± S.E.M of three experiments. *p* value was calculated versus control condition (0 mM); NS: not statistically significant.

## Data Availability

The data presented in this study are available in the article.

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
