# Peer review of "Alpha-Lipoic Acid Plays a Role in Endometriosis: New Evidence on Inflammasome-Mediated Interleukin Production, Cellular Adhesion and Invasion"

_molecules, 2021, doi:10.3390/molecules26020288_

Round 1
Reviewer 1 Report
The manuscript presented by Di Nicuolo et al. investigates the effects of alpha-lipoic acid (ALA) on endometriotic cells in vitro. Especially effects of ALA on ER-beta, NALP-3 and secretion of IL-1beta and IL-18 as well as adhesion, invasion and MMP activities were analyzed. All in all the connection between ER-beta and the inflammasome/inflammation remains controversial to me. As published by Tang et al. 2019 in Cells 8;1123 ER-beta can prevent inflammation, but in endometriosis it increases inflammation. Is there any hypothesis/observation why ER-beta acts so differently?
Major concerns
In the Abstract, Introduction and Discussion it was concluded that ALA might inhibit endometriosis progression or counteract alterations in endometriosis. However, ALA was only tested on cells and not on endometriotic lesions, thus, these conclusions are clearly misleading and exaggerated.
In the Introduction Ref 17 does not state anything about NALP-3 in endometriosis but instead deals with ER-beta and endometriosis.
In the Introduction as well as in the Discussion I wonder why the authors did not mention the inhibitory effects of ALA on NFkappaB or MMP9 (e.g. Shay et al. 2009; BBA 1790, 1149).
Materials and Methods
Page 4 Which dilutions of the antibodies (also actin) have been used? How was the protein transfer done (semi-dry)? Please mention the sensitivity and detection range of all ELISAs used. How many cells were used for the ELISAs in which plates and which media? Please provide more details about the experiments. How many cells and which media have been used for the Invasion assays and gelatin zymography?
Page 7 No dose-dependency of ALA on 12Z IL-1beta and 22B IL-18 could be observed, thus it remains unclear whether a specific effect could be observed.
Page 9 Again, no dose-dependency of ALA on 12Z ICAM-1 expression was found.
Discussion, page 10. The statement that ALA inhibited adhesion and invasion is too far-reaching, because you only showed a partial inhibition. It remains unclear to me why ALA should be important for endometriosis, because all experiments presented have been performed with endometriotic and not with endometrial cells. The process of endometriosis starts with endometrial cells from the uterus, which then travel to distant organs and then adhere and invade the tissue(s). Why did the authors not use primary endometrial cells to test the effects of ALA?
The statement that the authors showed how ALA is able to reduce ER-beta protein expression is somehow misleading, because no mechanism of reduction was shown.
Minor concerns
In the Introduction NALP-3 and NLRP3 are both used without making it clear that it is the same.
Page 3 the medium was changed or media were changed
Table 1 Please replace increasing concentration by increasing concentrations
Figs4/5 You labeled both Figures with A and B, but it was not explained in the Figure legend.
Author Response
Review 1
Thank You for Your comments and questions.
- The manuscript presented by Di Nicuolo et al. investigates the effects of alpha-lipoic acid (ALA) on endometriotic cells in vitro. Especially effects of ALA on ER-beta, NALP-3 and secretion of IL-1beta and IL-18 as well as adhesion, invasion and MMP activities were analyzed. All in all the connection between ER-beta and the inflammasome/inflammation remains controversial to me.
As published by Tang et al. 2019 in Cells 8;1123 ER-beta can prevent inflammation, but in endometriosis it increases inflammation. Is there any hypothesis/observation why ER-beta acts so differently?
Please consider the following references showing an inflammatory role of ERβ:
ERβ was shown to play a pro-inflammatory role in endometriosis. ERβ is expressed at many fold high levels in endometriotic tissue as compared to normal endometrium (1, 2). Over-expression of ERβ enhances the growth of endometriotic lesions in mice and interacts with components of the NLRP3 inflammosome such as NALP3, caspase-1, and caspase 9 and induces the processing of pro-IL-1β to its active form (3). In addition, several other inflammatory mediators such as macrophage inflammatory protein 1α, macrophage inflammatory protein 2, IL-16, monocyte chemoattractant protein 5, B lymphocyte chemoattractant, were upregulated upon ERβ over-expression in endometrial cells (1).
Conversely, there are several studies suggesting an anti-inflammatory role for ERβ in other contexts. For example, ERβ activation by selective agonists was shown to suppress TNFα induced gene expression by recruiting the co-activator SRC-2 forming a repressive complex (4). ERβ activation suppresses NF-κB activation and secretion of multiple cytokines in prostate cancer cell lines (5).
Thus, the ramifications of ERβ activation on immune activation should be analyzed separately in each disease context, keeping the possibility of ligand specific effects in mind, to determine the net effects on immune activation.
Ref.
- Mal R, Magner A , David J, Datta J, Vallabhaneni M et al. Estrogen Receptor Beta (ERβ): A Ligand Activated Tumor Suppressor. Front. Oncol. 2020,10: 1-14
- Dyson MT, Bulun SE. Cutting SRC-1 down to size in endometriosis. Nat Med., 2012, 18:1016–8.
- Han SJ, Jung SY, Wu SP, Hawkins SM, Park MJ, Kyo S, et al. Estrogen receptor β modulates apoptosis complexes and the inflammasome to drive the pathogenesis of endometriosis. Cell. 2015, 163: 960–74.
- Cvoro A, Tatomer D, TeeM-K, Zogovic T, Harris HA, Leitman DC. Selective estrogen receptor-β agonists repress transcription of proinflammatory genes. J Immunol. 2008, 180:630–6.
- Xiao L, Luo Y, Tai R, Zhang N. Estrogen receptor β suppresses inflammation and the progression of prostate cancer. Mol Med Rep. 2019. 19:3555–63.
Major concerns
- In the Abstract, Introduction and Discussion it was concluded that ALA might inhibit endometriosis progression or counteract alterations in endometriosis. However, ALA was only tested on cells and not on endometriotic lesions, thus, these conclusions are clearly misleading and exaggerated.
Thank you for your comment. A limit of the present study might be that we used human immortalized endometriotic cell lines from peritoneal red lesions. Epithelial (12Z) and stromal (22B) cells were derived from active red peritoneal endometriosis lesions during the proliferative phase of the menstrual cycle from woman suffering from endometriosis (1). Both cell lines share several phenotypic and molecular characteristics of primary cultured endometriotic cells and several data indicate that 12Z and 22B cells are a potential model system to study the progressive phase of endometriosis (1-4). Importantly, xenograft of a mixed population of these 12Z and 22B cells into the peritoneal cavity of immunocompromised mice is able to proliferate, attach, invade, reorganize and establish peritoneal endometriosis-like lesions and that histomorphology is similar to that of spontaneous peritoneal endometriosis in women (1,5).
We agree that we have to extend to endometriotic tissue the use of ALA. We have modified the sentences in the discussion according your suggestion:
In conclusion our findings provided new evidences: ALA has the ability to inhibit ER-β expression, NALP-3-induced cytokines production, migration and invasion in endometriotic cell lines.
We studied only human immortalized endometriotic cell lines derived from peritoneal lesions. Both cell lines share several phenotypic and molecular characteristics of primary cultured endometriotic cells and several data indicate that 12Z and 22B cells are a potential model system to study the progressive phase of endometriosis. Therefore, more studies using primary stromal and epithelial cells isolated from endometriotic tissues, are needed to confirm protective effect of ALA on endometriosis progression.
Ref:
- Zeitvogel, A., Baumann, R. and Starzinski-Powitz, A. Identification of an invasive, N-cadherin-expressing epithelial cell type in endometriosis using a new cell culture model. Am J Pathol 2001, 159: 1839-52.
- Banu, S.K., Lee, J., Speights Jr, V.O., Starzinski-Powitz, A. and Arosh, J.A. Selective inhibition of prostaglandin E2 receptors EP2 and EP4 induces apoptosis of human endometriotic cells through suppression of ERK1/2, AKT, NFkB and b-catenin pathways and activation of intrinsic apoptotic mechanisms. Molecular Endocrinology 2009, 23: 1291-1305.
- Kochunov, P., Glahn, D.C., Fox, P.T., Lancaster, J.L., Saleem, K., et al. Genetics of primary cerebral gyrification: Heritability of length, depth and area of primary sulci in an extended pedigree of Papio baboons. Neuroimage 2010, 53:1126-34.
- Lee, J., Banu, S.K., Subbarao, T., Starzinski-Powitz, A. and Arosh, J.A. Selective inhibition of prostaglandin E2 receptors EP2 and EP4 inhibits invasion of human immortalized endometriotic epithelial and stromal cells through suppression of metalloproteinases. Mol Cell Endocrinol 2011 332: 306-13.
- Arosh, J.A., Lee, J., Balasubbramanian, D., Stanley, J.A., Long, C.R., Meagher, M.W., et al.. Molecular and preclinical basis to inhibit PGE2 receptors EP2 and EP4 as a novel nonsteroidal therapy for endometriosis. Proc Natl Acad Sci U S A 2015,112: 9716-21.
- In the Introduction Ref 17 does not state anything about NALP-3 in endometriosis but instead deals with ER-beta and endometriosis.
Thank you for the correction, we have changed the reference.
3)In the Introduction as well as in the Discussion I wonder why the authors did not mention the inhibitory effects of ALA on NF-kappaB or MMP9 (e.g. Shay et al. 2009; BBA 1790, 1149).
Thank you for the comment, we added both in the introduction and in the discussion the data on the inhibitory effect of ALA on MMP-9 and NF-kB.
Materials and Methods
Thank you for the suggestion. We have included this information in the text:
Page 4 Which dilutions of the antibodies (also actin) have been used?
Anti- NALP3 human antibody monoclonal mouse IgG,Clone CL0210, Novus Biologicals, 1:1000;
anti –ERβ, Human ERβ/NR3A2 Antibody Monoclonal Mouse IgG1 Clone 733930, R&D Systems, 1:1000;
anti-human β-actin antibody, Sigma-Aldrich, 1:5000.
How was the protein transfer done (semi-dry)?
We use wet transfer method for blotting, 120 volts for 1hr.
Please mention the sensitivity and detection range of all ELISAs used.
Thank you for the suggestion. We add this information in the text:
IL-18: Detection Range 15.6-1000 pg/ml. Sensitivity: The minimum detectable dose of IL-18 is typically less than 5.9 pg/ml.
IL-1β: Detection Range 15.6-1000 pg/ml. Sensitivity: The minimum detectable dose of IL-1β is typically less than 5.7 pg/ml.
ICAM: Detection Range 15.6-1000 pg/mL. Sensitivity: The minimum detectable dose of this kit is typically less than 5.9 pg/mL.
How many cells were used for the ELISAs in which plates and which media?
IL-18 and IL-1β levels were measured in supernatants obtained from 12Z and 22B cells (1x105 well/ 24 well plate) treated with ALA (0-10 mM) for 24 hrs by an enzyme-linked immunoassay (ELISA).
I-CAM levels were measured in total lysates obtained from 12Z and 22B cells (2x105 well/ 24 well plate) treated with ALA (0-10 mM) for 24 hrs, by an enzyme-linked immunoassay (ELISA).
Culture media: Dulbecco's modified Eagle's medium (DMEM)/F12 medium supplemented with 10% fetal bovine serum, 100 U/ml penicillin G and streptomycin (LifeTechnologies,NY, U.S.A.) in a humidified atmosphere of 5% CO2–95% air at 37 °C.
Please provide more details about the experiments. How many cells and which media have been used for the Invasion assays and gelatin zymography?
The 96 Well 3D Spheroid BME Cell Invasion Assay utilizes a 3D Culture Qualified 96 Well Spheroid Formation Plate alongside a specialized Spheroid Formation ECM to drive aggregation and/or spheroid formation of cells. In our experiments, spheroids of 12Z or 22B cells (3x103/50 µl of cell suspension per well) were incubated with ALA (0-10 mM) for six days.
For gelatin zymography, cells (2x105 well/ 24 well plate) in serum free (DMEM)/F12 medium were treated with ALA (0-10 mM) for 24 hrs.
Page 7 No dose-dependency of ALA on 12Z IL-1beta and 22B IL-18 could be observed, thus it remains unclear whether a specific effect could be observed.
Page 9 Again, no dose-dependency of ALA on 12Z ICAM-1 expression was found.
Thank you for your comments. We observed a significant reduction in the 12Z IL-1β and 22B IL-18 secretion and on 12Z ICAM-1 expression after 24 hrs of ALA treatment, but this was not dose dependent. Probably, in our in vitro model, ALA reaches its maximum effect already at the dose of 1.0 mM.
Discussion, page 10. The statement that ALA inhibited adhesion and invasion is too far-reaching, because you only showed a partial inhibition. It remains unclear to me why ALA should be important for endometriosis, because all experiments presented have been performed with endometriotic and not with endometrial cells. The process of endometriosis starts with endometrial cells from the uterus, which then travel to distant organs and then adhere and invade the tissue(s). Why did the authors not use primary endometrial cells to test the effects of ALA? The statement that the authors showed how ALA is able to reduce ER-beta protein expression is somehow misleading, because no mechanism of reduction was shown.
Thank you so much for your comment. It is well known the use of human immortalized cell lines instead of cells obtained from fresh tissues. The use is also justified by the greater number of cells available for analysis. We totally agree with your suggestion and we will plan our next experiments on cells obtained from endometrial and endometriotic tissues.
Minor concerns
In the Introduction NALP-3 and NLRP3 are both used without making it clear that it is the same.
Page 3 the medium was changed or media were changed
Table 1 Please replace increasing concentration by increasing concentrations
Figs4/5 You labeled both Figures with A and B, but it was not explained in the Figure legend.
Thank you so much for your comments. We made these changes during the revision of the manuscript.

Reviewer 2 Report
Journal: Molecules
Title: Alpha-lipoic acid plays a role in endometriosis: new evidences on inflammasome-mediated interleukin production, cellular adhesion and invasion.
Comments:
In this manuscript, the authors describe the potential effect of α-lipoic acid (ALA) on ER-β expression, on NALP-3 –induced cytokines production, on migration and invasion of immortalized epithelial (12 Z) and stromal endometriotic cells (22B). The research is important in the approached field since endometriosis is frequently diagnosed in women and the mechanisms implicated in the pathogenesis of endometriosis may be of great interest for therapeutic purposes (i.e., identifying potentially novel molecular targets to cure such a disease). Overall, the article is well written, the research has been carefully conducted and the results are clearly presented, additionally containing suggestive (micro)photographs. The authors are applying actual techniques for the assessment of the analyzed parameters on 12 Z and 22B cell lines. Only a few inaccuracies were identified throughout the manuscript (see below):
- Abstract: I suggest adding in the abstract a few details regarding the utilized methods, and maybe reducing the background information in the abstract;
- Discussion (second paragraph): please add ”et al” after Han în the following sentence - Han has recently demonstrated that the ER-β …..
Author Response
Review 2
Comments:
In this manuscript, the authors describe the potential effect of α-lipoic acid (ALA) on ER-β expression, on NALP-3 –induced cytokines production, on migration and invasion of immortalized epithelial (12 Z) and stromal endometriotic cells (22B). The research is important in the approached field since endometriosis is frequently diagnosed in women and the mechanisms implicated in the pathogenesis of endometriosis may be of great interest for therapeutic purposes (i.e., identifying potentially novel molecular targets to cure such a disease). Overall, the article is well written, the research has been carefully conducted and the results are clearly presented, additionally containing suggestive (micro)photographs. The authors are applying actual techniques for the assessment of the analyzed parameters on 12 Z and 22B cell lines. Only a few inaccuracies were identified throughout the manuscript (see below):
- Abstract: I suggest adding in the abstract a few details regarding the utilized methods, and maybe reducing the background information in the abstract;
- Discussion (second paragraph): please add ”et al” after Han în the following sentence - Han has recently demonstrated that the ER-β …..
Thank You, we have modified the abstract and the sentence in the discussion as you suggest.
